# Direct probing of phonon mode specific electron–phonon scatterings in two-dimensional semiconductor transition metal dichalcogenides

Duk Hyun Lee[1,3], Sang-Jun Choi [2,3], Hakseong Kim[1], Yong-Sung Kim[1] & Suyong Jung [1✉]

Electron–phonon scatterings in solid-state systems are pivotal processes in determining many key physical quantities such as charge carrier mobilities and thermal conductivities. Here, we report direct probing of phonon mode specific electron–phonon scatterings in layered semiconducting transition metal dichalcogenides $WSe_2$, $MoSe_2$, $WS_2$, and $MoS_2$ through inelastic electron tunneling spectroscopy measurements, quantum transport simulations, and density functional calculation. We experimentally and theoretically characterize momentum-conserving single- and two-phonon electron–phonon scatterings involving up to as many as eight individual phonon modes in mono- and bilayer films, among which transverse, longitudinal acoustic and optical, and flexural optical phonons play significant roles in quantum charge flows. Moreover, the layer-number sensitive higher-order inelastic electron–phonon scatterings, which are confirmed to be generic in all four semiconducting layers, can be attributed to differing electronic structures, symmetry, and quantum interference effects during the scattering processes in the ultrathin semiconducting films.

[1] Korea Research Institute of Standards and Science, Daejeon, Republic of Korea. [2] Institute for Theoretical Physics and Astrophysics, University of Wűrzburg, Wűrzburg, Germany. [3] These authors contributed equally: Duk Hyun Lee, Sang-Jun Choi. ✉email: syjung@kriss.re.kr

The collective vibrational modes in atomically arranged structures, namely phonons, and their interactions with charged carriers play crucial roles in determining various properties of condensed matter systems, covering thermal capacity and conductivity, electron mobility, and superconductivity, to name a few[1-3]. Specifically, in two-dimensional (2D) van der Waals (vdW) layered materials, electron–phonon interactions have been widely recognized as key elements in characterizing electronic, optical, and quantum properties[4,5]. Among those, electron–phonon scatterings are considered to determine the intrinsic charge carrier mobility in 2D semiconducting transition metal dichalcogenides (SC-TMDs)[6-11]. For instance, as schematically illustrated in Fig. 1a, two-phonon modes $E''$ and $A_2''$ are optically accessible but weakly interact with electrons. In comparison, $E'$ and $A_1$ phonons are Raman active while strongly interacting with electrons. Transverse and longitudinal acoustic phonons have been regarded as the primary factors that limit in-plane charge carrier mobilities in 2D vdW SC-TMD films[8,12]. Direct experimental approaches to explore electron–phonon interactions, however, have been largely missing, and most previous reports on phonon-related phenomena have been limited to either one or two isolated phonons and their temporal interactions with optically pumped hot electrons, which would not be relevant in active electronic applications[13,14].

Free from stringent optical selection rules, inelastic electron tunneling spectroscopy (IETS) is known for its effectiveness in detecting electron–phonon interactions with a high-energy resolution[15-19]. Previously, scanning tunneling microscopy (STM) has been used for local IETS measurements of a few phonon excitations in 2D semimetals[16,17,20,21]. For instance, van Hove singularities of graphene phonon bands and phonon-mediated inelastic channels to graphene have been observed under an STM probe[16,17]. When it comes to 2D SC-TMDs, however, IETS studies with local probes have been sparse due to weak tunnel signals from the point-like metallic probe. IETS measurements through the atomically sharp tunnel junctions additionally suffer from the limitation of broad momentum spectra, making it further daunting to isolate phonon-mode specific electron–phonon scatterings from concerted scattering networks on the Fermi surface of the 2D materials.

In this work, we report IETS measurements with four prototypical type-VI 2D SC-TMD films as tunnel media and characterize single- and two-phonon IETS features involving up to as many as eight distinctive phonon modes in mono- and bilayer films. We find out that the momentum-conserving electron–phonon scattering processes in 2D semiconductors, reflecting individual phonon-mode specific electron–phonon coupling strengths, are governed by layer-number dependent electronic structures and inversion symmetry. Moreover, we identify that several two-phonon inelastic electron tunneling processes differ between the mono- and bilayer SC-TMD films. Corroborated with quantum transport simulation and density

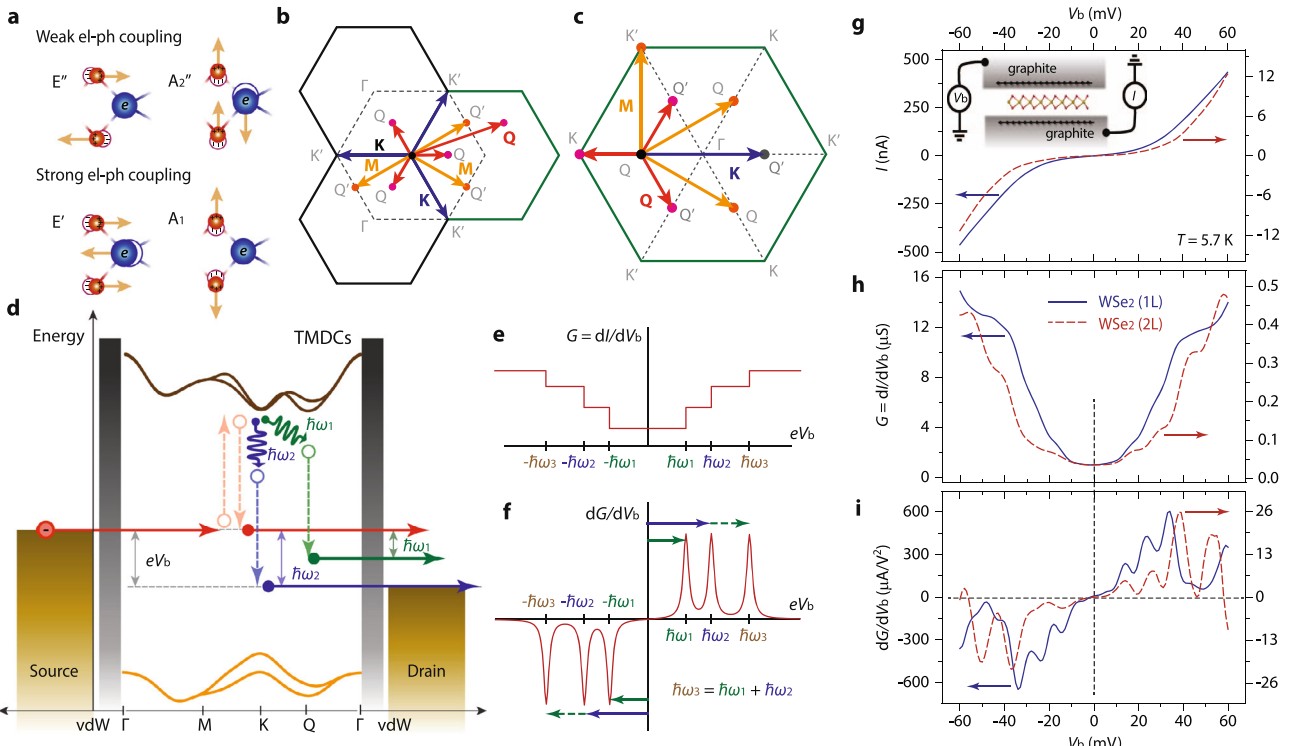

**Fig. 1 Inelastic electron tunneling spectroscopy probing electron–phonon scatterings in SC-TMD films. a** Schematic viewgraphs of lattice vibration modes that are either weakly (upper) or strongly (lower) coupled to electrons in 2H-SC-TMD layers. **b, c** Schematic illustrations of electron–phonon scatterings initiated at either the K (**b**) or Q (**c**) valley in the momentum space of a 2D hexagonal lattice. **d** Schematic of the energy-band alignment and quantum tunneling events in a vertical graphite–SC-TMD–graphite planar tunnel junction. Injected electrons within an energy window of $eV_b$ either elastically tunnel (red arrows) through the barriers or interact with SC-TMD phonons without (blue arrows) or with (green arrows) a momentum transfer before exiting to a drain electrode. **e, f** Simplified viewgraphs of the evolutions of differential conductance ($G = dI/dV_b$, **e**) and a second derivative of tunnel current ($dG/dV_b = d^2I/dV_b^2$, **f**) as a function of sample-bias voltage ($V_b$). Experimental signatures of the single- and two-phonon electron–phonon couplings are represented as a $dI/dV_b$ increase (**e**) and a $dG/dV_b$ peak or dip (**f**) at individual phonon energies $\hbar\omega_1$ and $\hbar\omega_2$ and combined phonon-mode energy $\hbar\omega_3 = \hbar\omega_1 + \hbar\omega_2$. **g–i** Inelastic electron tunnel spectra from a mono- (solid blue lines) and bilayer (dotted red line) WSe$_2$ planar tunnel junction at $T = 5.7$ K. Tunnel electron characteristics are represented in $I$ – $V_b$ (**g**), $dI/dV_b$ – $V_b$ (**h**), and $dG/dV_b$ – $V_b$ (**i**) curves, respectively. (inset, **g**) Schematic of our graphite–SC-TMD–graphite vertical planar junction and two-probe electric measurement setup.

functional perturbation theory (DFPT), we suggest that the layer-number sensitive electron–phonon scatterings can be understood by a quantum interference effect and symmetry-regulated geometric phase in the higher-order scattering processes.

## Results

**IETS with 2D vdW junctions.** A simplified device scheme is illustrated in the inset of Fig. 1g. We implement thin graphite (>5 nm) as source and drain electrodes to preserve the intrinsic physical and electrical properties of the mono- and bilayer SC-TMDs, while introducing vdW tunnel barriers at the 2D vertical heterojunctions[22–24]. Another advantage of utilizing graphite as contact electrodes is to minimize the momentum mismatch of the electrons tunneling to and from the 2H-SC-TMDs, which enables efficient monitoring of the electron–phonon scattering processes around the K (Fig. 1b) and Q (Fig. 1c) points, in which the conduction band edges of the hexagonal 2D semiconducting layers are located[24]. As schematically illustrated in Fig. 1d, electrons injected through a barrier transiently excite to the conduction band edges of the insulator. Although a majority of the electrons elastically tunnel through the insulator while releasing the pre-borrowed excitation energy (dashed and solid red lines in Fig. 1d), electron–phonon interactions allow some electrons to be scattered off to other electronic states with or without a momentum change (solid blue and green wavy arrows in Fig. 1d, respectively). Accordingly, adjunct transport channels are constructively established in the charge flows through the tunnel junctions, through which inelastic tunneling events are exhibited as conductance modulations ($G = dI/dV_b$, Fig. 1e) or as peaks or dips of the second derivative of tunnel current ($dG/dV_b$, Fig. 1f) depending on sample bias ($V_b$) polarities. During these inelastic electron tunneling processes, momentum conservations limit the accessible phonons to the excitations at specific high-symmetric points, so that phonon excitations probed by our inelastic quantum tunneling measurements can be directly associated with the elemental electron–phonon scattering processes at the high-symmetric points and essentially govern the charge flows through the tunnel media. In that regard, our IETS spectra cannot simply be related to the combined phonon density of states of the SC-TMDs, for which the detailed information of phonon momentum is obscured.

Figure 1g–i shows a collection of inelastic electron tunneling spectra from the first set of mono- (solid blue lines) and bilayer (dotted red lines) WSe₂ planar tunnel junctions at $T = 5.7$ K. We add an AC excitation voltage ($V_{pp} = 1$ mV, $f = 43.33$ Hz) to DC $V_b$ and simultaneously measure both $I$–$V_b$ (Fig. 1g) and $G = dI/dV_b$–$V_b$ (Fig. 1h) with an AC lock-in amplifier[19]. Within the $V_b$ range of $|V_b| \leq 100$ mV, vertical charge flows through our graphite–WSe₂–graphite planar junctions are governed by direct tunneling, which is sensitive to WSe₂ layer thickness, tunnel junction area, crystallographic misalignment angles between the graphite electrodes and WSe₂ tunnel insulator, and electron–phonon scatterings. As explained before, weak IETS signals in $dI/dV_b$ (Fig. 1e and h) can be better displayed as peaks and valleys in the second derivative of the tunnel current, $dG/dV_b = d^2I/dV_b^2$ (Fig. 1f and i), with their symmetric locations around $V_b = 0$ mV representing the characteristic IETS feature linked to electron scatterings with phonons of activation energy $eV_b$[15]. It is worth pointing out that $dG/dV_b$ spectra shapes are sensitive to the tunnel-junction parameters, so we limit our discussions to the devices with weak couplings, free from strongly coupled resonance tunneling[25]. We obtained the $dG/dV_b$ spectra by numerically differentiating $G = dI/dV_b$, confirmed to be consistent with data independently measured from an additional AC lock-in amplifier synchronized at a frequency of $2f$ (Supplementary Fig. 1).

The conspicuous $dG/dV_b$ signals indicate that tunnel electrons heavily interact with WSe₂ phonons, while the differing $dG/dV_b$ values and $V_b$ positions between the mono- and bilayer devices imply that electron–phonon scatterings in the 2D vdW heterostructures are WSe₂ layer-number variant. We note that all IETS spectra presented in the manuscript are obtained by numerically averaging out as many as 121 individual $dG/dV_b$ spectra, often varying external gate voltage $V_g$ applied to the Si/SiO₂/h-BN back gate (Supplementary Fig. 2). By following this procedure, we can identify $V_g$-invariant IETS signals as peaks or dips, while the $V_g$-dependent features, such as those relating to defect states and the electronic structures of the SC-TMDs and graphite electrodes, can be avoided[19].

**Single-phonon electron scatterings in mono- and bilayer WSe₂.** Figures 2b and c respectively show comprehensive IETS spectra from the second set of mono- and bilayer WSe₂ devices, measured at $T = 0.45$ K with an excitation voltage of $V_{pp} = 0.3$ mV. In total, we are able to identify eight independent IETS features in both mono- and bilayer samples within an energy window of $|eV_b| \leq 38$ meV and then compare them with DFPT-calculated phonon dispersions of freestanding monolayer WSe₂ (Fig. 2a) and graphene–WSe₂–graphene heterostructures (Supplementary Fig. 3). The $V_b$ positions for each IETS feature are assessed through a multi-peak Lorentzian fitting (solid grey curves in Fig. 2b), with the collective fitting result (overlaid orange line in Fig. 2b) matching the experimental data very well. Moreover, the open blue squares in Fig. 2b and c respectively indicate the experimental data in negative $V_b$ from the mono- and bilayer devices, with the $dG/dV_b$ dip locations perfectly aligning with the $dG/dV_b$ peaks in positive $V_b$. The widths of the ticks (Fig. 2b and c) indicate the intrinsic IETS spectra-broadening in our measurements: FWHM ≈ 0.6 meV [26].

Now we undertake the identification of the phonon modes responsible for each IETS feature, based on theoretical reports on the electron–phonon coupling strengths in 2D semiconductors[6–11]. Since momentum-conserving virtual quantum tunneling primarily occurs at the K and Q valleys, high-symmetric phonons connecting the K and Q valleys can be expected to contribute strongly in the electron–phonon scattering processes in monolayer WSe₂. First, we mark the $dG/dV_b$ peak labeled as P₂ (8.57 mV ± 0.18 mV) in Fig. 2b is associated with the TA-phonon branch, or more specifically, TA (Q) phonon excitation, which is the high-symmetry phonon mode exclusively excited within our measurement uncertainty. Similarly, P₃ (13.71 mV ± 0.07 mV) in the monolayer junction can be assigned to the LA(Q) phonons, which are predicted to cause strong electron–phonon interactions[6–11]. The next signal, P₄ (15.95 mV ± 0.05 mV), is within the phonon excitation energies of LA(K) and LA (M). As a note, transverse acoustic (ZA) phonons are expected to deform electrostatic potentials weakly, thus making electron scatterings with ZA phonons irrelevant in our measurements. It is worth mentioning that the IETS features marked with P₁ in the monolayer (2.36 mV ± 0.12 mV) and bilayer (3.80 mV ± 0.13 mV) can be attributed to a newly excited lattice vibration mode in the graphite–WSe₂–graphite heterojunctions. Such low-energy excitations can be related to the intricate crystallographic arrangements of 2D vdW interfaces[19].

Among the other six optical phonon branches of WSe₂ monolayers, LO₂, TO₂, and ZO₁ phonon modes are expected to cause strong electron–phonon scatterings[6–11]. Accordingly, with relative ease, we can assign P₅ (24.63 mV ± 0.08 mV) in the monolayer to the LO₂(K) phonon. Identifying the primary phonon excitations for the next three $dG/dV_b$ peaks, however, is not as straightforward as those for the previously assigned low-energy phonons. For example, P₆ (27.48 mV ± 0.04 mV) is in close proximity to LO₂(M)/TO₂(M) and LO₂(Q), while P₇ (31.33

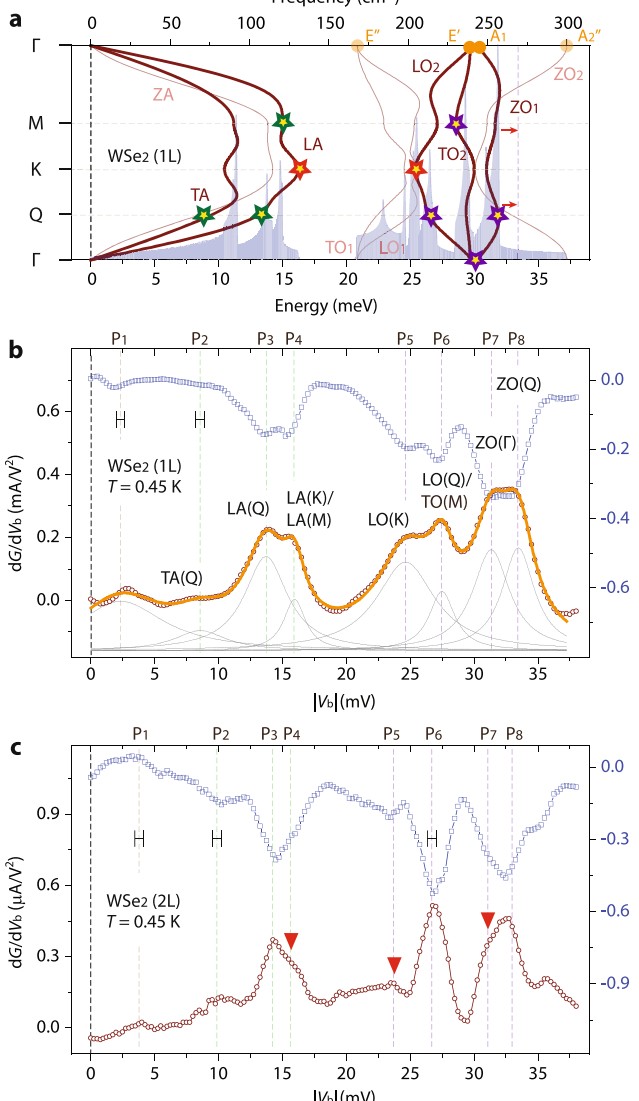

**Fig. 2 Single-phonon electron–phonon scatterings in mono- and bilayer WSe₂. a** DFPT-calculated phonon dispersion and phonon density of states of freestanding monolayer WSe₂. The phonon branches generating strong electron–phonon couplings, namely TA, LA, LO₂, TO₂, and ZO₁, are indicated with bold lines, and the optically accessible phonon modes at Γ, namely E″, E′, A₁, and A₂″, are marked with orange dots. Colored stars mark the experimentally verified phonon excitations that strongly couple to electrons. **b, c** dG/dV_b evolutions of the positive (open brown circles) and negative (open blue squares) V_b in mono- (**b**) and bilayer (**c**) WSe₂ tunnel devices. The IETS spectra were taken at T = 0.45 K with an excitation voltage of V_pp = 0.3 mV.

tunneling at the K (K′) valley irrelevant because the interlayer couplings at the K (K′) points are weak around the conduction band edges[27–29]. Instead, strong interlayer hybridizations are prompted by the orbitals responsible for the conduction band edges at Q. Therefore, in bilayer WSe₂, electrons tunneled into the first layer from the graphite electrode with momentum K (K′) should be scattered off to Q (Q′) through intra- or intervalley electron–phonon scatterings (Fig. 1b) before tunneling to the second layer[27,28]. Accordingly, single-phonon interlayer tunneling assisted by K phonons becomes limited in SC-TMD bilayers, leading to diminished inelastic tunnel features with the K phonons. As marked with red inverted triangles in Fig. 2c, the electron–phonon scattering signal with LO₂(K) phonons is indeed reduced in intensity in the bilayer (P₅ in Fig. 2c) when compared with the conspicuous IETS feature for LO₂(K) phonons in the monolayer (Fig. 2b). Similarly, the dG/dV_b hump marked as P₄ (15.60 mV ± 0.28 mV) becomes attenuated in intensity, and the moderate undertone of the dG/dV_b spectra, P₇ in the bilayer, could be attributed to quenched TO₂(K) and ZO₁(K) phonon excitations as well.

**Two-phonon electron scatterings in mono- and bilayer WSe₂.** Thanks to an excellent tunnel-junction stability, we are able to probe high-energy inelastic electron tunneling processes in 2D SC-TMDs. Figures 3a and b respectively show dG/dV_b spectra from the mono- and bilayer WSe₂ devices up to |eV_b| ≤ 70 meV. Note that no available WSe₂ phonon modes exist within the energy range 40 meV ≤ |eV_b| ≤ 70 meV, with the majority of phonons associated with the graphite having much higher energies[19,30]. Interestingly, these high-energy IETS spectra differ by eV_b location between the mono- and bilayer devices. In the monolayer, for example, a distinct dG/dV_b peak is observed at ≈ 59 meV along with a rather broad dG/dV_b hump at ≈53 meV. Meanwhile, as shown in Fig. 3b for the bilayer, the strongest IETS signal is formed at ≈42 meV, at which no apparent dG/dV_b spectra exist in the monolayer WSe₂. In addition, the high-energy spectra in the bilayer (40 meV ≤ |eV_b| ≤ 60 meV) are far stronger in intensity than the IETS signals at low energies (|eV_b| ≤ 40 meV), which is not the case in the monolayer device.

We accredit the sets of higher-energy layer-number-dependent IETS spectra to two-phonon electron–phonon scatterings that can be interpreted through quantum interference and the geometric phase in ultrathin WSe₂ films. Let us begin with an intuitive description of quantum interference and the geometric phase in two-phonon electron–phonon scatterings, during which an electron with an initial momentum $k_i$ is interacting with two respective phonons of momenta $q$ and $q'$ and ends up in a state with a final momentum of $k_f = k_i - q - q'$. In a microscopic view, momentum conservation allows two different electron–phonon inelastic processes with differing scattering orders: emitting the first $q(q')$ phonon and stopping at the intermediate state $\kappa_A = k_i - q(\kappa_B = k_i - q')$, and arriving at $\kappa_f$ after scattering off by emitting the second $q'(q)$ phonon. Note that such a pair of distinctive scattering routes form a closed loop in momentum space, and the developed quantum superposition finally determines the inelastic electron tunneling probability that is responsible for the experimentally observable IETS signals in magnitude (insets in Fig. 3c and d). Quite notably, owing to the peculiar electronic structures of the SC-TMDs with six distinct Q and K valleys around the conduction band edges, such a quantum interference effect comes into play for various two-phonon electron–phonon scattering processes. For instance, when scattered by M and Q phonons such as LA(M) and LO(Q), an electron at the K point is allowed to travel through two intermediate states ($\kappa_A$, $\kappa_B$) at Q₁ and Q₄ before arriving at the K′ point (inset in Fig. 3d). During these scattering processes, the quantum interference effect

mV ± 0.07 mV) and P₈ (33.45 mV ± 0.06 mV) are within the energy ranges of the TO₂ (Γ, K)/ZO₁(Γ, K) and ZO₁(Q, M) phonon excitations. Here, we point out that identifying the primary optical modes from these energetically close-packed WSe₂ optical phonons can be possible with higher-order two-phonon IETS measurements and quantum transport simulations, as we discuss in detail later.

We find that the IETS spectra from the bilayer device, thus the electron–phonon scattering characteristics in WSe₂ bilayers, are distinct when compared with their monolayer counterparts, which we relate to the layer-number variant symmetry and the electronic structures around the conducting channels. Inversion symmetry preserved in bilayer 2H SC-TMDs renders interlayer

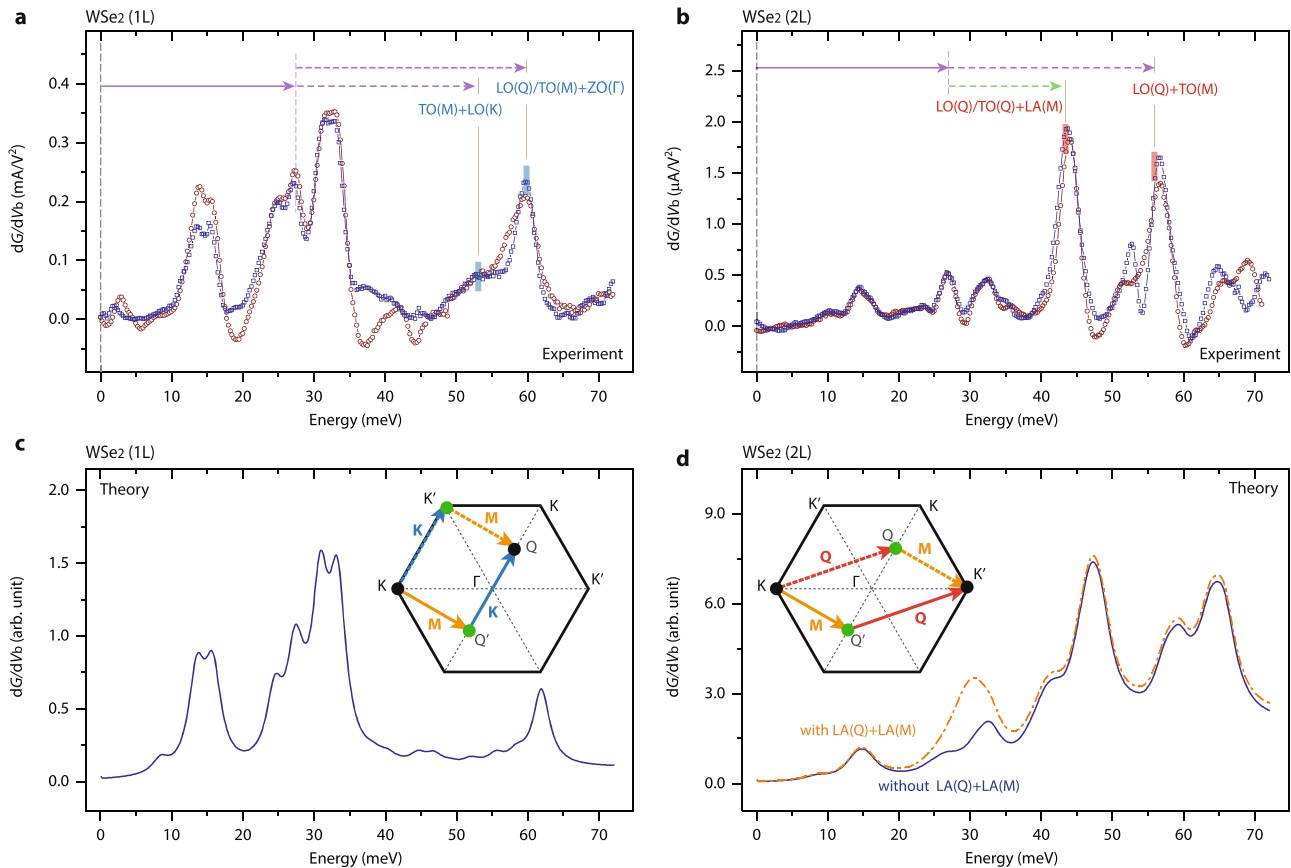

**Fig. 3 Two-phonon electron–phonon scatterings in mono- and bilayer WSe₂. a**, **b** Detailed d$G$/d$V_b$ evolutions of positive (open brown circles) and negative (open blue squares) $V_b$ in mono- (**a**) and bilayer (**b**) WSe₂ tunnel devices. The IETS spectra were taken at $T = 0.45$ K with an excitation voltage of $V_{pp} = 0.3$ mV. The lengths of the colored arrows, used for guiding the two-phonon excitations, represent the primary phonon energies in the monolayer WSe₂. **c**, **d** Simulated d$G$/d$V_b$ spectra from quantum transport calculations for mono- (**c**) and bilayer (**d**) WSe₂ films. The single-phonon excitations are based on the monolayer data (Fig. 2b), and the two-phonon electron–phonon scatterings are from the composite two-phonon scattering routes regulated by layer-number dependent electronic band structures, inversion symmetries, and geometric phases. (insets, **c**, **d**) Schematic illustrations of two-phonon electron–phonon scattering routes at the K valley interacting with M and K phonons (**c**), and M and Q phonons (**d**).

becomes prominent since the probability of each scattering route is the same thanks to the identical band structures around the $Q_1$ and $Q_4$ points. In stark contrast, however, when tunnel electrons are scattered by the phonons at M and K, during which the electrons detour through the intermediate states at the K and Q valleys (inset in Fig. 3c), the quantum interference effect becomes attenuated due to the dissimilar tunneling probabilities resulting from the differing energy gaps at the K and Q valleys.

When a sudden spin change accompanies the course of electrons traveling through a closed loop in momentum space, moreover, the geometric phase can play a pivotal role in determining the quantum interference[31–35]. In the monolayer SC-TMDs, for instance, where time-reversal symmetry is preserved but inversion symmetry is not, the additional geometric phase π is added in the two-phonon inelastic electron scattering processes with M and Q phonons. On the other hand, the simultaneous presence of time-reversal and inversion symmetries in the SC-TMD bilayers forces the geometric phase to vanish, and accordingly, the quantum phase around the closed loop becomes equivalently 2π, resulting in constructive quantum interference in the bilayer films (Supplementary Note)[36,37]. We further note that higher-order electron–phonon scatterings and their experimental realizations in the systems where a spin-momentum locking is absent could be simply related to the joint density of states of interacting phonons. However, in the SC-TMDs where the spin-momentum locking

is present, the quantum interference and the geometric phase come as major players in the two-phonon electron scattering processes (Supplementary Note).

With a rigorous quantum mechanical IETS simulations (see Methods and the Supplementary Note for detailed descriptions), we show that the higher-energy two-phonon IETS signals are indeed sensitive to the aforementioned quantum interference and geometric phase. We consider all possible combinations of two-phonon inelastic scattering routes out of the experimentally identified eight individual phonons, and confirm that the two-phonon IETS signals specifically associated with Q and M phonons are sensitive to the layer-number dependent symmetries and quantum interference. We figure vertical charge flows through the bilayer as interlayer tunneling through two WSe₂ films with the exclusion of intervalley scattering around the K(K′) in the scattering matrix. Figure 3c and d displays simulated d$G$/d$V_b$ spectra for mono- and bilayer WSe₂ with full consideration of the geometric phase and quantum interference around the closed loop, and the agreement with the experimental data is eminent–in particular, the absence (presence) of d$G$/d$V_b$ signals within $40$ meV $\leq |eV_b| \leq 55$ meV in the mono- (bi) layer WSe₂ films. We provide all the theoretically expected positions of IETS features and two-phonon inelastic scattering processes exhibiting quantum interference in Supplementary Table 1. It should be pointed out that our transport model for bilayer WSe₂ has shortcomings to fully analyze the experimental observations. For

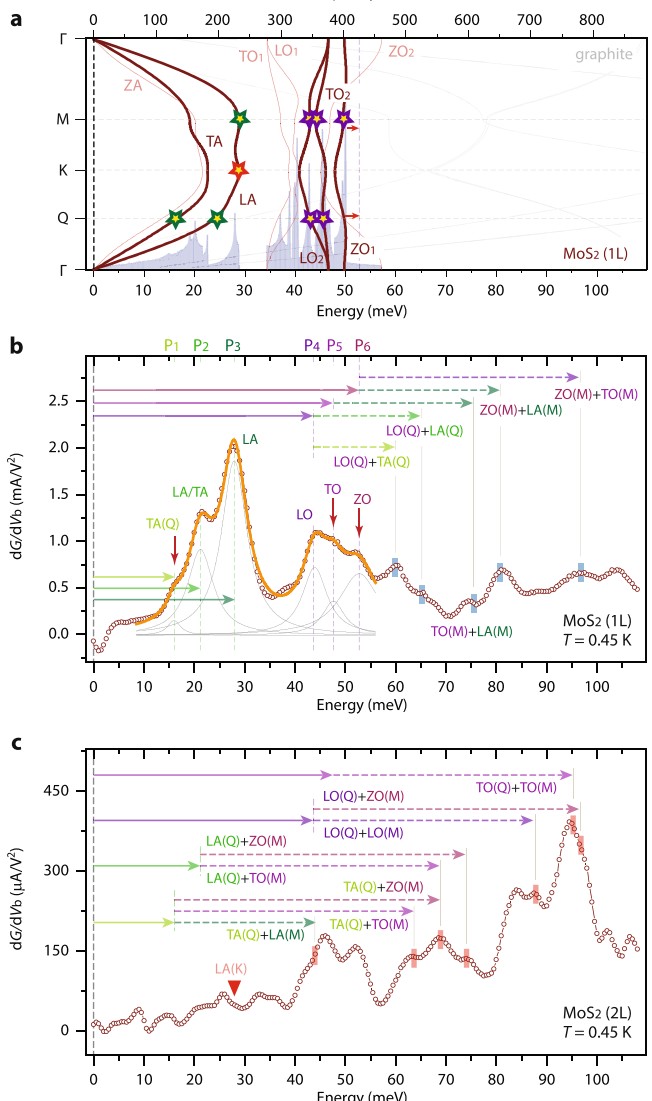

**Fig. 4 Single- and two-phonon electron–phonon scatterings in mono- and bilayer MoS₂. a** DFPT-calculated phonon dispersion and phonon density of states of freestanding monolayer MoS₂. The graphite phonon branches are illustrated with thin grey lines. The phonon branches confirmed to generate strong electron–phonon scatterings are indicated with bold lines. **b**, **c** d*G*/d*V*$_b$ evolutions as a function of energy in mono- (**b**) and bilayer (**c**) MoS₂ tunnel devices. The IETS spectra were taken at *T* = 0.45 K with an excitation voltage of *V*$_{pp}$ = 0.5 mV. The experimentally identified single phonon-mode energies are distinctly marked with solid and dotted colored arrows, and the two-phonon electron–phonon scattering features are indicated with vertical blue (**b**) and red (**c**) tick marks, guided with vertical grey lines.

example, the two-phonon inelastic electron scatterings with LA phonons are expected to largely contribute to the vertical charge flows, resulting in higher d*G*/d*V*$_b$ spectra in value around 25 meV ≤ |*eV*$_b$| ≤ 35 meV (dotted orange line in Fig. 3d). Instead, the simulated spectra without considering the double LA phonon scatterings (solid blue line in Fig. 3d) are found to be closer to the experimental observations, calling for further theoretical works to better clarify the phonon interactions with conducting electrons in bilayer WSe₂.

**Electron–phonon scatterings in MoS₂, MoSe₂, and WS₂.** We find out that the key electron–phonon scattering characteristics observed with WSe₂ films—that single- and two-phonon electron–phonon scatterings are regulated by layer-number dependent electronic structures and symmetries—are generic to the other type-VI SC-TMD films, namely MoS₂, MoSe₂, and WS₂. Figure 4b shows an IETS spectrum from a monolayer MoS₂ device, measured at *T* = 0.45 K with an excitation voltage of *V*$_{pp}$ = 0.5 mV. From the monolayer MoS₂, we can identify six distinct IETS features within an energy window of |*eV*$_b$| ≤ 55 meV, with each spectrum closely aligned to the TA, LA, LO₂, TO₂, and ZO₁ phonon branches (Fig. 4a). Identical to the monolayer WSe₂, TA(Q), LA(Q), and LA(M, K) phonons can be marked as the leading acoustic phonon excitations that generate sizable electron–phonon scatterings in monolayer MoS₂. Separated by a sizable gap at 30 meV ≤ |*eV*$_b$| ≤ 45 meV, three d*G*/d*V*$_b$ peaks appear close together and can be linked to LO₂, TO₂, and ZO₁ optical phonons. As discussed previously, the IETS spectra, thus electron–phonon scatterings in bilayer MoS₂ films contrast those in monolayer MoS₂. Figure 4c shows an IETS spectrum from a bilayer MoS₂ device, and the IETS feature corresponding to the single-phonon electron scatterings with LA(K) phonons becomes diminished in the bilayer MoS₂, as marked with an inverted red triangle in Fig. 4c. Moreover, the high-energy IETS features at 55 meV ≤ |*eV*$_b$| ≤ 100 meV, which are attributed to the two-phonon inelastic scatterings, are higher in intensity than those for the low-energy IETS signals. The vertical red tick marks and colored solid and dotted arrows in Fig. 4c indicate various two-phonon inelastic electrons scattering processes with Q and M phonons. In comparison, as marked with vertical blue tick marks in Fig. 4b for the monolayer, most high-energy d*G*/d*V*$_b$ peaks can be explained by the two-phonon electron–phonon scatterings, save for the destructive Q and M combinations. Based on these observations, we can infer that the elementary phonon modes in ultrathin MoS₂ are TA(Q), LA(Q), LA(M), LO₂(Q)/LO₂(M), TO₂(Q)/TO₂(M), and ZO₁(M) phonons.

Figures 5c and e respectively display IETS measurements from the mono- and bilayer MoSe₂ tunnel devices, and Fig. 5d and f are those from the WS₂ mono- and bilayer devices, along with the DFPT-calculated phonon dispersions and phonon density of states of freestanding monolayer MoSe₂ (Fig. 5a) and WS₂ (Fig. 5b). The tunnel spectra observed in MoSe₂ and WS₂ are consistent with the previously discussed electron–phonon scattering physics in WSe₂ and MoS₂, in particular, inversion symmetry-regulated charged carrier scattering with K phonons: LA(K) as indicated with inverted red triangles in Fig. 5e and f, and geometric phase administered Q and M two-phonon inelastic scatterings. Some of the prominent two-phonon modes for MoSe₂ and WS₂ are marked with colored arrows and their combinations in Fig. 5c–f, with the primary single-phonon excitations identified in our measurements denoted with colored stars in Fig. 5a for MoSe₂ and Fig. 5b for WS₂. It is interesting to point out that, unlike WSe₂, MoS₂, and MoSe₂, the most prominent d*G*/d*V*$_b$ peak in bilayer WS₂ is located at low energy ≈32 meV because a sizable phonon gap is formed between the acoustic and optical phonon branches; we can attribute such a strong peak to two-phonon electron scatterings with acoustic phonons of TA(Q) + LA(M) and TA(M) + LA(Q). In total, we measured four mono- and four bilayer WSe₂ devices, with all showing consistent single- and two-phonon electron–phonon scattering features (Supplementary Figs. 4 and 5). Our findings were additionally confirmed with three mono- and two bilayer MoSe₂, two mono- and three bilayer WS₂, and one mono- and one bilayer MoS₂ planar tunnel junctions. IETS spectra for the devices not discussed in the main text are presented in Supplementary Figs. 6 and 7.

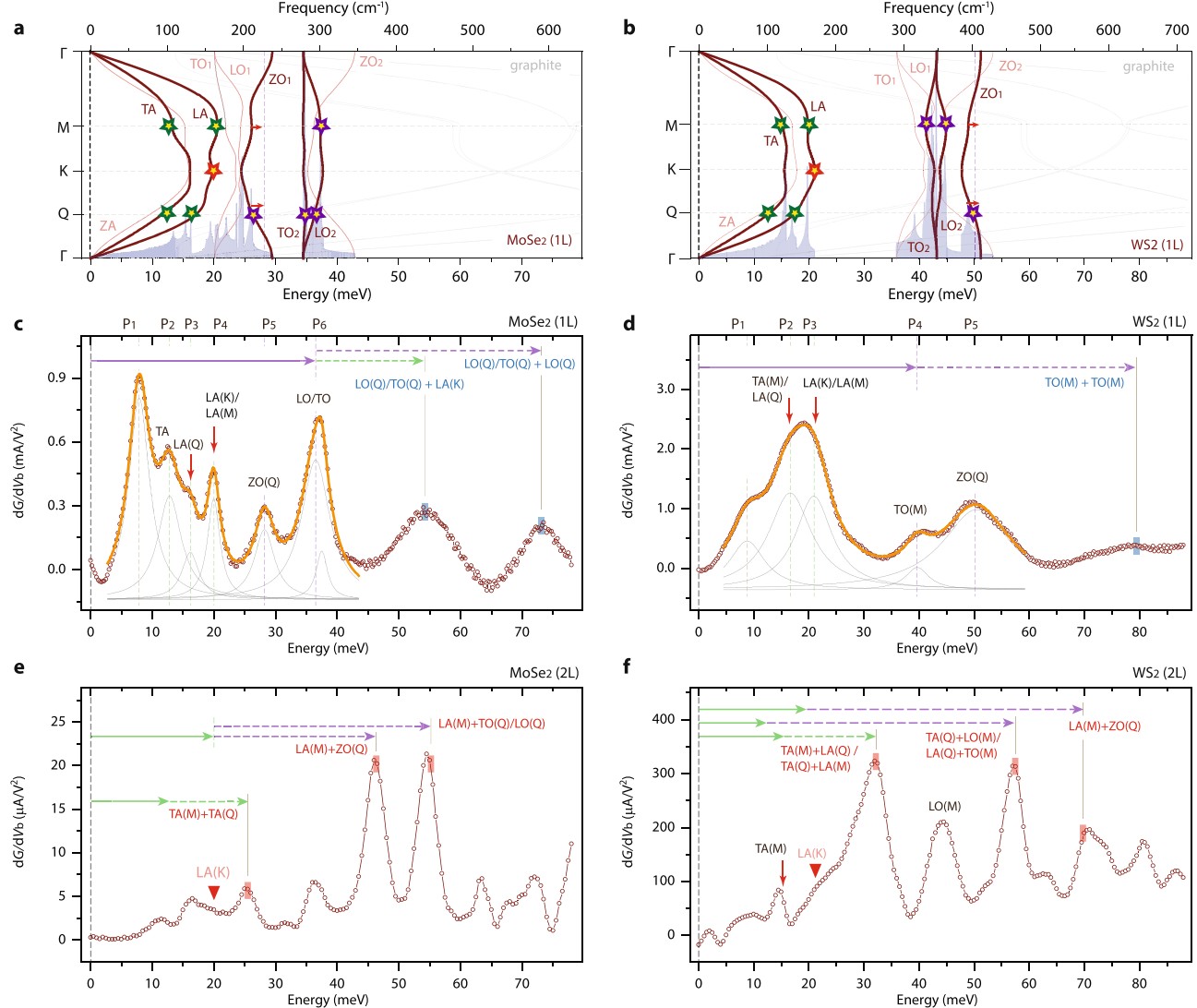

**Fig. 5 Electron–phonon scatterings in mono- and bilayer MoSe2 and WS₂. a, b** DFPT-calculated phonon dispersions and phonon density of states of freestanding monolayer MoSe$_2$ (**a**) and WS$_2$ (**b**). **c–f** d$G$/d$V_b$ evolutions as a function of energy in mono- and bilayer MoSe$_2$ and WS$_2$ tunnel devices. The IETS spectra were taken at $T = 0.45$ K with an excitation voltage of $V_{pp} = 0.5$ mV. The experimentally identified single phonon-mode energies are marked with solid and dotted colored arrows, and the two-phonon electron–phonon scatterings are indicated with vertical tick marks.

Lastly, we remark that the experimentally identified ZO$_1$ phonons are consistently higher in energy than the theoretically expected freestanding SC-TMD phonon excitations, as indicated with red arrows in Fig. 2a (WSe$_2$), Fig. 4a (MoS$_2$), Fig. 5a (MoSe$_2$), and Fig. 5b (WS$_2$), suggesting that flexural motions of chalcogen atoms become hardened in 2D vdW vertical heterostructures by as much as $\approx 3$ meV[19]. Although it is not straightforward to draw the exact phonon dispersions of our graphene–SC-TMD–graphene heterojunctions, primarily due to the lattice mismatches, we discover that the assessable ZO phonon density of states indeed shifts to higher energies, even in the simplest graphene($4 \times 4$)–WSe$_2$($3 \times 3$) heterostructures (Supplementary Fig. 3).

## Discussion

To further support our findings, we prepared another type of graphite–SC-TMD–graphite vertical heterostructure: twisted double-layer WSe$_2$ vertical junctions. As shown in the optical image in Supplementary Fig. 8, two monolayer WSe$_2$ films are serially transferred on top of the bottom graphite, and the misalignment angle of the double WSe$_2$ layers is estimated to be

around 13° as judged from the crystallographic directions of each layer. Distinct from Bernal stacked bilayers, the inversion symmetry of the twisted double-layer WSe$_2$ is naturally broken such that the IETS features from the double-layer device should be quite dissimilar to those from conventional bilayer WSe$_2$. Indeed, we find that the overall IETS signals in the double-layer device are similar to those from the monolayer films in terms of the excited phonon modes, without noticeable high-energy d$G$/d$V_b$ features that could relate to the Q + M two-phonon excitations (Supplementary Fig. 8). Although more in-depth experimental works should follow to clarify the compelling electron–phonon scatterings in twisted double-layer systems, we feel confident that the current data sufficiently support our interpretation made in the current manuscript: layer-number variant electronic structures, symmetry, and quantum interference play important roles in both single- and two-phonon electron–phonon scattering processes in SC-TMD films.

As previously remarked, we implement graphite flakes as the source and drain electrodes in our vertical planar tunnel junctions to preserve the intrinsic electronic properties of the ultrathin SC-TMD layers. As electrons tunnel through the graphite–SC-TMD

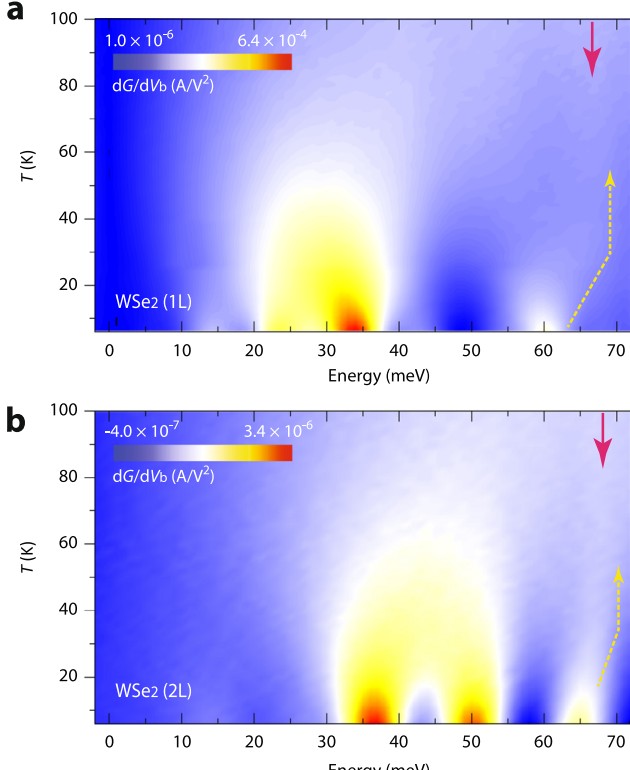

**Fig. 6 Temperature-dependent IETS spectra in mono- and bilayer WSe$_2$.** **a**, **b** Two-dimensional display of d$G$/d$V_b$ evolutions of the first set of mono- (**a**) and bilayer (**b**) WSe$_2$ tunnel devices as a function of $V_b$ at varying temperatures from $T = 6$ K to $T = 100$ K with $\Delta T = 2$ K. Dotted yellow and magenta arrows indicate the $T$-dependent evolutions of the WSe$_2$ two-phonon electron–phonon scatterings to the graphite ZO(K) phonons at $\approx 66$ meV.

vertical junctions, therefore, there is a high chance they will be scattered by graphite phonons as well. From our IETS measurements, we are indeed able to locate several d$G$/d$V_b$ spectra that are likely related to the graphite phonons and the two-phonon inelastic electron scatterings with the phonons of the SC-TMDs and graphite layers. For example, the d$G$/d$V_b$ spectra at $\approx 66$ meV in both mono- and bilayer devices, as respectively marked with red arrows and dotted yellow lines in Fig. 6a and b, are from the graphite ZO(K) phonons[16,18,19]. It is worth mentioning that the graphite ZO(K) signals persist up to $T \geq 100$ K, in stark contrast to the temperature sensitive two-phonon IETS features discussed above. Moreover, the graphite phonon modes emerge only when the two-phonon WSe$_2$ electron–phonon scatterings become diminished at elevated temperatures ($T > 30$ K). We are also able to observe several other graphite and two-phonon graphite–SC-TMD phonon signatures within the energy range $|eV_b| \leq 300$ meV, but detailed analyses of these high-energy spectra are beyond the current study and will be presented elsewhere. Finally, moving beyond bilayer, we find that the IETS signals originating from higher-order electron–phonon scatterings in three- and four-layer SC-TMD devices are consistently higher in intensity, as presented in Supplementary Fig. 9, suggesting that charge flows through multi-layered SC-TMDs become heavily regulated and are often facilitated by multiple electron–phonon scattering processes.

In summary, we spectroscopically characterized phonon-mode specific electron–phonon scatterings in four prototypical 2D semiconducting films, WSe$_2$, MoS$_2$, WS$_2$, and MoSe$_2$, by IETS measurements, quantum transport simulations, and density

functional theory. Thanks to the solid physical and electrical stability of the planar tunnel junctions, we were able to probe several single- and two-phonon inelastic electron scattering processes that are sensitive to the layer-number dependent electronic structures, symmetry, and geometric phase. From the standpoint of electron–phonon scattering physics in condensed matter systems, our experimental measurements suggest that quantum interference can be a major player in momentum-conserving inelastic electron–phonon scatterings and thus charge transport behaviors in 2D SC-TMDs. In addition, we demonstrated that our experimental approach, utilizing inelastic tunneling spectroscopy with 2D planar vdW tunnel junctions as a high-fidelity material metrology platform, is applicable to a wide range of low-dimensional quantum materials and their unlimited combinations for probing charge carrier interactions with phonons and other intriguing quasiparticles[38,39].

## Methods

**Device fabrication.** In our planar vdW heterostructures, preparation of atomically clean interfaces is of critical importance for an accurate and reliable material characterization of the vertical junctions. At first, 60–100 nm thick h-BN flakes are mechanically exfoliated on a 90 nm thick SiO$_2$ layer on Si substrate. Then, a mechanically isolated graphite flake of thickness 5 nm or more is transferred to a pre-located h-BN flake on the SiO$_2$/Si substrate using a dry transfer method. We utilize polymer stacks of PMMA (poly(methyl methacrylate))–PSS (polystyrene sulfonate) layers for such tasks and carefully adjust the thickness of each layer to enhance the optical contrast of the exfoliated ultrathin 2D layered materials. We remove the PMMA film in warm (60 °C) acetone and further anneal the samples at 350 °C for several hours in a mixture of Ar:H$_2$ = 9:1 to ensure residue-free graphite surfaces. Next, instead of again using the polymer stacks, we use a Gel-Pak to exfoliate and transfer mono- and bilayer SC-TMD films on top of the h-BN–graphite stack. Gel-Pak residue-free surfaces are confirmed with an atomic force microscope measurement. Finally, a top graphite flake prepared on the PMMA–PSS polymer stack is transferred to form a vertical graphite–SC-TMD–graphite planar tunnel device. We note that the crystallographic angles of the graphite electrodes and the SC-TMD films are intentionally misaligned, and the thicknesses of the semiconducting layers are confirmed via atomic force microscope. The active junction areas, which are determined by the widths of the top and bottom graphite flakes, are several tenths of a square micrometer. We purchased high-purity (>99.995%) SC-TMD crystals from HQ Graphene with no additional dopants added during growth procedures and large size graphenium flakes from NGS Naturgraphit GmbH.

**DFPT for calculating phonon dispersion.** Phonon dispersions of free-standing SC-TMD monolayers and graphene–WSe$_2$–graphene heterostructures are calculated using DFPT, implemented in Vienna Ab initio Simulation Package[40] within generalized gradient approximation (PBE)[41]. Projector augmented pseudopotentials are used and the plane-wave cutoff is set to be 500 eV[42]. The phonon dispersions of free-standing SC-TMD monolayers are calculated with a 3 × 3 supercell and 2 × 2 k-point mesh. The phonon structures of graphene–WSe$_2$–graphene heterostructures are calculated with a 3 × 3 WSe$_2$ supercell and 4 × 4 graphene supercell with 2 × 2 k-point mesh as well. The lattice constant of WSe$_2$ is set to be 3.325 Å in PBE, and the graphene lattices are relaxed by –1% to compensate for any WSe$_2$–graphene lattice mismatch.

**Quantum transport simulation.** Quantum transport simulations are performed using an electron-tunneling scattering matrix with two-particle Green functions. The two-particle Green function $G_{\kappa\kappa'}(\tau, s, t)$ determines the transmission probability of the tunnel junctions in the time domain as follows

$$T(\epsilon_f, \epsilon_i) = \sum_{\kappa,\kappa'} \int \int \int d\tau ds dt \, e^{\frac{i[(\epsilon_i - \epsilon_f)\tau + \epsilon_f t - \epsilon_i s]}{\hbar}} G_{\kappa\kappa'}(\tau, s, t), \quad (1)$$

where $\tau, s, t > 0$. The transmission probability $T(\epsilon_f, \epsilon_i)$ is used to calculate the vertical electron tunneling current, $I(V) \propto \int d\epsilon_f d\epsilon_i T(\epsilon_f, \epsilon_i)[f_L(\epsilon_i - eV) - f_R(\epsilon_f)]$. The Green function consists of the product of a probability amplitude and its complex conjugation, which correspond to the propagation of electrons moving forward and backward in the time domain. Although algebraic evaluation of the two-particle Green function is complicated, Feynman diagrams as provided in Fig. 7 simplify our calculations while providing intuitive understanding. Each vertex represents electron–phonon interaction with coupling strength $M_{\kappa',\kappa}^\lambda$. Here, $\lambda$ denotes a phonon mode that scatters an electron from momentum state $\kappa$ to $\kappa'$. When electrons scatter with the phonon modes $\lambda_r$ and $\lambda_b$, the momentum-conserving two-phonon inelastic electron scatterings allow four independent scattering processes, as depicted in the Feynman diagram (Fig. 7). Then, the transmission probability $T$ can be estimated with the absolute square of the sum of

**Fig. 7 Feynman diagrams of $G^2_{\kappa\kappa'}(\tau,s,t)$ for the second-order peaks of dG/dV$_b$.** The solid lines with right and left arrows represent the propagations of electrons moving forward and backward in the time domain in the conduction band, respectively. The backward propagation appears as a complex conjugation of the probability amplitude in the forward propagation. Colored wavy lines denote the exchange of phonons between electrons and holes, while different colors refer to different phonon modes such as LA and LO. Two different tunneling processes via intermediate states of $\kappa_A$ and $\kappa_B$ exhibit quantum interference when those processes involve two phonons with a reversed order. The interaction strength of each term is presented before the parentheses. We highlight that quantum interference occurs as the intermediate states $\kappa_A$ and $\kappa_B$ are distinct in 2D SC-TMDs.

the two electron–phonon scattering amplitudes, i.e., $T = |A+B|^2 = |A|^2 + |B|^2 + A(B)^* + (A)^*B$, where the first and second terms represent the first and second Feynman diagrams in Fig. 7. The third and fourth terms, which are responsible for the quantum interference, represent the third and fourth Feynman diagrams, respectively. Detailed evaluations of $G_{\kappa\kappa'}(\tau, s, t)$, Hamiltonian, and geometric phase are presented in the Supplementary Information.

## Data availability

All data supporting the findings of this study are available from the corresponding authors on request.

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

## Acknowledgements

We thank Y.-W.S. and H.S.S. for their careful reading and comments on our manuscript. This work was supported by research grants for basic research (KRISS-2020-GP20011059) funded by the Korea Research Institute of Standards and Science and the Basic Science Research Program (NRF-2019R1A2C2004007) through the National Research Foundation of Korea. This work was also supported by the DFG (SFB1170 "ToCoTronics"), the Würzburg-Dresden Cluster of Excellence ct.qmat, EXC2147, project-id 39085490.

## Author contributions

S.J. and S-J.C. designed the experiments, and D.H.L and H.K. fabricated the devices and performed the inelastic electron tunneling spectroscopy measurements. S-J.C. carried out the quantum transport simulations and Y-S.K. performed the DFPT calculations. D.H.L., S-J.C., Y-S.K., and S.J. analyzed the data and co-wrote the paper. All authors contributed to the manuscript.

## Competing interests

The authors declare no competing interests.
