## [Peer Review File · Nature Communications]

Reviewers' Comments:

Reviewer #1:
None

Reviewer #2:
None

Reviewer #3:
None

Reviewer #4:
Remarks to the Author:
Dear Editors,

I have checked the replies to the questions raised by the reviewers and believe that the authors have properly attempted to address these concerns. Since the reviewer maintains that these effects are not like the authors describe, I suggest the authors in the interest of scientific progress to rephrase their statements in a more cautious language. For instance: When they write [..]Moreover, we identify several two-phonon inelastic electron tunneling processes that are governed by layer-number dependent symmetry, quantum interference, and the Berry phase. [] they could instead state that these processes are interpreted in terms of quantum interference and Berry phase.

There are several other instances, where a more cautionary language would alleviate concerns and leave the ultimate interpretation somewhat open.

A further example on p9 in which the discussion turns to Berry phase, the authors could describe their analysis based on this hypothesis.

Having said that, the authors should then also add a brief discussion into their paper, which could contain pretty much the statements made to the reviewers. I believe this could be of interest to the community also.

Reviewer(s)' Remarks to the Author:

Reviewer #4

Comments:

I have checked the replies to the questions raised by the reviewers and believe that the authors have properly attempted to address these concerns. Since the reviewer maintains that these effects are not like the authors describe, I suggest the authors in the interest of scientific progress to rephrase their statements in a more cautious language. For instance: When they write [...] Moreover, we identify several two-phonon inelastic electron tunneling processes that are governed by layer-number dependent symmetry, quantum interference, and the Berry phase. [] they could instead state that these processes are interpreted in terms of quantum interference and Berry phase. There are several other instances, where a more cautionary language would alleviate concerns and leave the ultimate interpretation somewhat open.

A further example on p9 in which the discussion turns to Berry phase, the authors could describe their analysis based on this hypothesis.

Having said that, the authors should then also add a brief discussion into their paper, which could contain pretty much the statements made to the reviewers. I believe this could be of interest to the community also.

Response:

We thank the reviewer for positive comments on the status of our current manuscript, and deeply appreciate their thoughtful recommendations to alleviate any impending concerns on our data interpretation. Following their suggestions, we have carefully revised the manuscript covering the sections of abstract, introduction, main result and discussion with more cautionary language. For instance, we have replaced the terminology of ‘Berry phase’ with more general phrasing ‘geometric phase’ throughout the manuscript. We have also modified the title of our manuscript, excluding ‘Symmetry and Berry phase’ in the prevision submission. For your convenience, we highlight all the revisions and corrections in red in the revised manuscript.

Complying with the reviewer’s suggestion, we have also included a brief discussion on the geometric phase and quantum interference effects in spin-momentum locked 2D semiconducting layers in the main text, and more detailed discussions including the ones made in reviewing processes have been added in the Supplementary Note.

Added in the main text (page 10):

..., the quantum interference effect becomes attenuated due to the dissimilar tunneling probabilities resulting from the differing energy gaps at the K and Q valleys.

When a sudden spin change accompanies the course of electrons traveling through a closed loop in momentum space, moreover, the geometric phase can play a pivotal role in determining the quantum interference.^{31–35} In the monolayer SC-TMDs, for instance, where time-reversal symmetry is preserved but inversion symmetry is not, the additional geometric phase π is added in the two-phonon inelastic electron scattering processes with M and Q phonons. On the other hand, the simultaneous presence of time-reversal and inversion symmetries in the SC-TMD bilayers forces the geometric phase to vanish, and accordingly, the quantum phase around the closed loop becomes equivalently 2π , resulting in constructive quantum interference in the bilayer films (Supplementary Note).^{36,37} We further note that higher-order electron–phonon scatterings and their experimental realizations in the systems where a spin-momentum locking is absent could be simply related to the joint density of states of interacting phonons. However, in the SC-TMDs where the spin-momentum locking is present, the quantum interference and the geometric phase come as major players in the two-phonon electron scattering processes (Supplementary Note).

With a rigorous quantum mechanical IETS simulations (see Methods and the Supplementary Note for detailed descriptions), we show that the higher-energy two-phonon IETS signals are indeed sensitive to the aforementioned quantum interference and geometric phase. ...